# Polysomnographic Evidence of Enhanced Sleep Quality with Adaptive Thermal Regulation

**DOI:** 10.3390/healthcare13192521

**Published:** 2025-10-04

**Authors:** Jeong-Whun Kim, Sungjin Heo, Dongheon Lee, Joonki Hong, Donghyuk Yang, Sungeun Moon

**Affiliations:** 1Department of Otorhinolaryngology-Head and Neck Surgery, Seoul National University College of Medicine, Seoul National University Bundang Hospital, 82 Goomiro 173 Boengil, Bundang-gu, Seongnam 13620, Republic of Korea; 2Asleep Research Institute, 431, Teheran-ro, Gangnam-gu, Seoul 06159, Republic of Korea; chris.heo@asleep.ai (S.H.); sleep@asleep.ai (D.L.); nocturne@asleep.ai (J.H.); lloyd.yang@asleep.ai (D.Y.); tessa.moon@asleep.ai (S.M.)

**Keywords:** sleep quality, thermoregulation, polysomnography, mattresses, sleep architecture

## Abstract

**Background/Objective:** Sleep is a vital determinant of human health, where both its quantity and quality directly impact physical and mental well-being. Thermoregulation plays a pivotal role in sleep quality, as the body’s ability to regulate temperature varies across different sleep stages. This study examines the effects of a novel real-time temperature adjustment (RTA) mattress, which dynamically modulates temperature to align with sleep stage transitions, compared to constant temperature control (CTC). Through polysomnographic (PSG) assessments, this study evaluates how adaptive thermal regulation influences sleep architecture, aiming to identify its potential for optimizing restorative sleep. **Methods:** A prospective longitudinal cohort study involving 25 participants (13 males and 12 females; mean age: 39.7 years) evaluated sleep quality across three conditions: natural sleep (Control), CTC (33 °C constant mattress temperature), and RTA (temperature dynamically adjusted: 30 °C during REM sleep; 33 °C during non-REM sleep). Each participant completed three polysomnography (PSG) sessions. Sleep metrics, including total sleep time (TST), sleep efficiency, wake after sleep onset (WASO), and sleep stage percentages, were assessed. Repeated-measures ANOVA and post hoc analyses were performed. **Results:** RTA significantly improved sleep quality metrics compared to Control and CTC. TST increased from 356.2 min (Control) to 383.2 min (RTA, *p* = 0.030), with sleep efficiency rising from 82.8% to 87.3% (*p* = 0.030). WASO decreased from 58.2 min (Control) and 64.6 min (CTC) to 49.0 min (RTA, *p* = 0.067). REM latency was notably reduced under RTA (110.4 min) compared to Control (141.8 min, *p* = 0.002). The REM sleep percentage increased under RTA (20.8%, *p* = 0.006), with significant subgroup-specific enhancements in males (*p* = 0.010). Females showed significant increases in deep sleep percentage under RTA (12.3%, *p* = 0.011). **Conclusions**: Adaptive thermal regulation enhances sleep quality by aligning mattress temperature with physiological needs during different sleep stages. These findings highlight the potential of RTA as a non-invasive intervention to optimize restorative sleep and promote overall well-being. Further research could explore long-term health benefits and broader applications.

## 1. Introduction

Sleep plays an essential role in maintaining human health and well-being. Insufficient or poor-quality sleep has been linked to a variety of negative health outcomes, including cognitive impairment, mood disorders, weakened immune function, and an increased risk of chronic diseases such as hypertension, diabetes, and obesity. While the quantity of sleep is important, research has increasingly focused on sleep quality, including factors such as sleep architecture, which refers to the structure and distribution of different sleep stages throughout the night [1,2]. Optimizing sleep architecture, which includes periods of light sleep, deep sleep, and rapid eye movement (REM) sleep, is crucial for restorative sleep that promotes both physical and mental recovery [3,4].

High-quality sleep enhances mental and physical health and aids in fatigue recovery. Several studies have evaluated the effects of environmental factors (e.g., temperature, humidity, airflow, light, and noise) on sleep quality [5,6,7,8]. Thermal environments, in particular, affect skin temperature, which varies dynamically across sleep stages (e.g., N1–3 and REM stages) [9]. Thermoneutrality, defined as the temperature range in which minimal metabolic heat production balances environmental heat loss, is critical for maintaining sleep quality. Extreme or sudden temperature changes outside this range are known to deteriorate sleep quality [10,11,12,13].

Maintaining thermoneutral conditions for the skin during sleep, which can be achieved through appropriate ambient and mattress temperatures, is essential for ensuring optimal sleep quality [13,14]. Although most studies focus on ambient temperature, the temperature of the mattress—which is closer to the body—has been found to exert a greater influence on sleep quality [9,12,14]. A stable and appropriate mattress temperature contributes significantly to achieving high-quality sleep, and certain studies have identified optimal ranges for mattress temperature to support this [15,16,17,18].

Various studies have explored the effects of environmental temperature on sleep quality. Many have demonstrated that sleeping in an environment that is too hot or too cold can disrupt sleep continuity, increase wakefulness, and shorten periods of REM and deep sleep, which are key for cognitive restoration and physical recovery [14,19,20]. Traditionally, thermal interventions have focused on controlling the ambient temperature of the room, but recent advancements in sleep technologies have enabled more localized temperature regulation, such as through heated blankets or temperature-adjustable mattresses. Most current approaches utilize a constant temperature throughout the night, which may not reflect the body’s dynamic thermoregulatory needs during different sleep stages [21,22,23].

This study aims to address this gap by investigating the effects of real-time mattress temperature adjustment (RTA) based on sleep stage transitions. Unlike constant temperature control (CTC), which maintains a fixed mattress temperature throughout the night, RTA dynamically adjusts the temperature in response to the sleep stage, thereby aligning the mattress temperature with the body’s changing thermoregulatory needs. Specifically, the temperature is lowered during REM sleep to reflect the body’s reduced ability to regulate temperature during this stage, while returning to a higher, more comfortable baseline during non-REM sleep. We hypothesize that this adaptive approach will result in improvements in sleep architecture, including increased total sleep time, sleep efficiency, and enhanced durations of REM and deep sleep, while reducing wakefulness after sleep onset (WASO).

## 2. Materials and Methods

### 2.1. Study Design and Participants

This study employed a prospective longitudinal cohort design to evaluate the effects of real-time temperature adjustment (RTA) on sleep quality compared to constant temperature control (CTC). Participants aged 19 to 60 years were recruited, ensuring equal representation across each decade of age. The eligibility criteria required participants to maintain consistent sleep–wake schedules for at least five days per week and to have no aversion to thermal mattress use. The exclusion criteria included the use of medical devices, prior surgery that could affect polysomnography (PSG) results, and requiring treatment for severe sleep disorders or mental health conditions, and the presence of moderate-to-severe obstructive sleep apnea (OSA), defined as an apnea–hypopnea index (AHI) ≥ 15 events/h. Recruitment was conducted via public advertisements, and all participants received detailed explanations of the study’s purpose, methods, and potential benefits before providing written informed consent. To minimize confounding influences on thermoregulation, the participants were instructed to avoid caffeine and alcohol on the day of each visit. In addition, they were asked to refrain from meals and vigorous exercise for at least 6 h prior to polysomnography. This study protocol was approved by the Institutional Review Board (IRB #P01-202401-01-042). Importantly, each participant underwent PSG on three separate occasions, allowing for robust within-subject comparisons under varying conditions.

### 2.2. Sleep Environment and Measurement Tools

The sleep environment was standardized to ensure consistency, with the room temperature maintained at 18–20 °C and humidity controlled at 50–55%. The participants adapted to the controlled sleep environment for approximately 1 h before the lights were turned off. This study was conducted during the winter months (December 2023 to February 2024) in Seoul, Republic of Korea, where the average outdoor temperature was approximately −2 °C, and relative humidity ranged from 55% to 74%. Sleep measurements were conducted using level-1 full-night PSG with the Nox A1 PSG system (Nox Medical, Reykjavik, Iceland), which complies with the standards of the American Academy of Sleep Medicine (AASM). PSG recordings included electroencephalography, electrooculography, chin and limb electromyography, electrocardiography, nasal pressure transducer, chest and abdomen respiratory inductance plethysmography, and pulse oximetry. Physiological data, including key sleep architecture metrics such as total sleep time, sleep efficiency, wake after sleep onset, and REM and deep sleep durations, were analyzed using the Noxturnal software version 7.1.1 (Nox Medical, Reykjavik, Iceland) and reviewed by a sleep expert.

The intervention used a thermal mattress (EMW720, KyungDong Navien, Seoul, Republic of Korea), with temperature regulated through an AI-based application. This application recorded participants’ breathing sounds during sleep and predicted sleep stages in real time using an AI model validated against PSG (macro F1 score of 0.76) [24,25]. The adaptive thermal regulation system dynamically adjusted mattress temperature based on transitions between sleep stages detected by the application.

### 2.3. Study Procedure for Thermal Mattress Temperature Control

The participants were randomly assigned to one of two groups to minimize potential order effects. They were informed that the mattress temperature might vary across experimental nights but were blinded to the specific study conditions and timing of temperature adjustments. Group 1 first underwent a baseline natural sleep condition without a mattress intervention, followed by the CTC condition and then the RTA condition. Group 2 followed this order in reverse, starting with natural sleep, then RTA, and concluding with CTC. Each condition was separated by a five-day interval to minimize residual effects and allow the participants to return to their baseline state (Figure 1).

Under baseline conditions, the participants slept without any mattress intervention. Under CTC conditions, the mattress temperature was maintained at a constant 33 °C throughout the night, reflecting the commonly preferred heating mat temperature in South Korea [26]. Under RTA conditions, the mattress temperature was dynamically adjusted during REM sleep and the temperature decreased by 3 °C to 30 °C to accommodate reduced thermoregulation, while during non-REM sleep it reverted to 33 °C [14,27]. Thirty minutes before the scheduled wake time, the temperature increased by 3 °C to 36 °C to facilitate arousal. (Figure 2) [14].

### 2.4. Statistical Analysis

Data from the three PSG sessions were analyzed to compare the effects of CTC and RTA conditions on sleep architecture. Repeated-measures ANOVA was conducted to evaluate differences across the conditions, followed by post hoc tests using the Friedman correction to identify specific variations. Statistical significance was defined as *p* < 0.05, and all analyses were performed using standard statistical software. The inclusion of three separate PSG sessions for each participant ensured a robust evaluation of the adaptive thermal regulation’s impact on sleep quality.

## 3. Results

### 3.1. General and Baseline Polysomnographic Characteristics

A total of 30 participants were initially recruited for this study. However, five participants were excluded from the final analysis due to their inability to complete all three polysomnographic (PSG) sessions. Therefore, data from 25 participants, consisting of 13 males and 12 females, were analyzed. The mean age of the participants was 39.7 years, with males averaging 42.4 years and females averaging 36.9 years (*p* = 0.336). The mean body mass index (BMI) was 25.2 kg/m^2^.

Participants with known moderate-to-severe obstructive sleep apnea (OSA) were excluded during recruitment. Baseline PSG analysis further confirmed that the mean apnea–hypopnea index (AHI) was within the non-clinically significant range, 5.48 events/h in males and 3.72 events/h in females, indicating that no participant had moderate-to-severe OSA.

Baseline polysomnographic (PSG) results revealed significant gender differences in sleep parameters. The average total sleep time (TST) was 356.2 min, with females exhibiting a non-significantly longer TST (372.6 min) compared to males (341.1 min) (*p* = 0.157). Sleep efficiency averaged 82.8%, with females achieving a non-significantly higher efficiency (86.6%) than males (79.3%) (*p* = 0.176). Wake after sleep onset (WASO) was longer in males (68.5 min) compared to females (45.3 min), with a group average of 58.2 min, demonstrating a statistically non-significant difference (*p* = 0.233). Sleep onset latency was also not significantly longer in males (21.0 min) compared to females (11.8 min) (*p* = 0.910), with a total group average of 16.3 min.

### 3.2. Comparative Analysis of Sleep Architecture Across Control, CTC, and RTA Conditions

This study evaluated sleep parameters under three conditions: natural sleep (Control), constant temperature control (CTC), and real-time temperature adjustment (RTA). The results are summarized in Table 1.

#### 3.2.1. Total Sleep Time and Sleep Efficiency

Significant improvements in both total sleep time (TST) and sleep efficiency were observed with the RTA condition. Total sleep time increased from 356.2 min under Control to 383.2 min under RTA (*p* = 0.030). Post hoc analysis revealed a statistically significant difference between Control and RTA (*p* = 0.025), while the difference between CTC and RTA (*p* = 0.054) approached significance. Similarly, sleep efficiency improved significantly from 82.8% under Control to 87.3% under RTA (*p* = 0.030). Post hoc comparisons indicated significant differences between Control and RTA (*p* = 0.004), as well as CTC and RTA (*p* = 0.010).

#### 3.2.2. Wake After Sleep Onset, Sleep Onset Latency, and Rem Latency

Wake after sleep onset (WASO) decreased under the RTA condition (49.0 min) compared to Control (58.2 min) and CTC (64.6 min), with a trend toward significance (*p* = 0.067). Post hoc analysis showed a significant reduction in WASO between CTC and RTA (*p* = 0.041).

Sleep onset latency showed no significant differences across the conditions (*p* = 0.383), with RTA maintaining a similar latency to CTC (8.3 vs. 8.4 min, respectively). REM latency significantly decreased under RTA (110.4 min) compared to Control (141.8 min) and CTC (113.5 min; *p* = 0.002). Additionally, post hoc analysis revealed significant reductions in REM latency between Control and RTA (*p* = 0.020).

#### 3.2.3. Sleep Stages

For light sleep, a significant reduction was observed across conditions (*p* = 0.002), with post hoc analysis showing a significant difference between Control and RTA (*p* = 0.011). No significant changes were observed in deep sleep duration (*p* = 0.482). REM sleep percentage showed an increasing trend from Control (17.7%) to RTA (20.8%; *p* = 0.006), although post hoc comparisons did not reveal significant differences among individual conditions (Figure 3).

### 3.3. Effects of Adaptive Thermal Regulation: Male and Female Subgroup Analysis

#### 3.3.1. Males (*n* = 13)

Subgroup analysis results are shown in Table 2. In the male subgroup, significant improvements in sleep efficiency and REM sleep percentage were observed with real-time temperature adjustment (RTA). Sleep efficiency increased from 79.3% under Control and 77.2% under constant temperature control (CTC) to 85.1% under RTA (*p* = 0.001), with post hoc analysis revealing significant differences between Control and RTA (*p* = 0.017) and CTC and RTA (*p* = 0.001). REM sleep percentage also significantly increased from 17.5% under Control to 21.3% under RTA (*p* = 0.010), although post hoc analysis did not show significant differences between the conditions.

A reduction in REM latency was observed under RTA (103.6 min) compared to Control (143.3 min) and CTC (112.3 min; *p* = 0.001). Post hoc analysis indicated a significant difference between Control and RTA (*p* = 0.018). Wake after sleep onset (WASO) and sleep onset latency showed trends toward improvement under RTA but did not reach statistical significance (*p* = 0.062 and *p* = 0.066, respectively).

Light sleep duration decreased under RTA (68.9%) compared to Control (76.5%) and CTC (74.1%), although this difference was not statistically significant (*p* = 0.170). Deep sleep percentage showed no significant differences among the conditions (*p* = 0.859) (Figure 4).

#### 3.3.2. Females (*n* = 12)

In the female subgroup, significant changes were observed in light sleep and deep sleep percentages. Light sleep decreased significantly under RTA (66.7%) compared to Control (75.1%) and CTC (69.8%; *p* = 0.001). Post hoc analysis showed a significant difference between Control and RTA (*p* = 0.035). Deep sleep percentage increased from 7.1% under Control to 12.3% under RTA (*p* = 0.011), although post hoc analysis did not reveal significant differences between the conditions.

Total sleep time (TST), sleep efficiency, WASO, sleep onset latency, and REM latency showed no significant differences across the conditions. REM sleep percentage exhibited a non-significant increase under RTA (20.3%) compared to Control (17.8%) and CTC (19.6%; *p* = 0.249) (Figure 4).

## 4. Discussion

The findings of this study provide compelling evidence that adaptive thermal regulation (RTA) significantly enhances sleep architecture compared to constant temperature control (CTC) and natural sleep. Improvements in total sleep time and sleep efficiency underscore the utility of dynamic temperature modulation in optimizing restorative sleep. By aligning mattress temperature with the body’s thermoregulatory needs during different sleep stages, RTA facilitates prolonged and uninterrupted sleep. These outcomes are consistent with prior studies that emphasize the role of environmental temperature in maintaining sleep continuity and reducing wakefulness after sleep onset [28,29]. RTA achieved a marked reduction in REM latency and an increase in REM sleep percentage, highlighting the critical role of temperature modulation during REM sleep in enhancing cognitive and emotional recovery processes [30]. Furthermore, the significant increase in deep sleep percentage under RTA supports the hypothesis that adaptive thermal regulation enhances the body’s recovery by optimizing thermoregulatory efficiency during non-REM sleep. These findings advocate for the utility of RTA in improving overall sleep quality and promoting restorative functions.

Most existing studies focus on identifying the optimal ranges for ambient and mattress temperatures, often relying on subjective measures such as surveys [19,31]. While these studies provide valuable insights, they largely overlook the dynamic temperature changes required during different sleep stages. This study differs from previous work by incorporating real-time temperature adjustments tailored to each sleep stage, accounting for physiological changes in body temperature [9,12,14]. Prior studies have identified the optimal mattress temperature for general sleep comfort as ranging between 28 °C and 31 °C [32], while some studies suggest higher limits, up to 35 °C, under specific conditions [26,33]. However, few studies have investigated the direct impact of mattress temperature adjustments on sleep stages. This study bridges this gap by assessing sleep quality under adaptive thermal conditions with advanced technologies, such as AI-based dynamic mattress temperature control.

This study identified gender-specific differences in response to RTA, highlighting the physiological and possibly hormonal variations that influence sleep architecture [34]. While males exhibited significant improvements in REM sleep percentage and efficiency, females benefited more from enhanced deep sleep duration. These differences may be attributed to distinct thermoregulatory profiles, as previous research suggests that hormonal fluctuations, such as those related to estrogen and progesterone levels, influence body temperature regulation and sleep patterns in females [35]. The reduced light sleep percentage and increased deep sleep observed in females under RTA suggest that this intervention could be tailored to meet gender-specific thermoregulatory needs. Moreover, the subgroup analysis revealed that males showed a greater reduction in REM latency under RTA, potentially reflecting a heightened sensitivity to the optimized thermal environment during REM sleep. These findings advocate for future research to explore personalized thermal interventions based on individual physiological characteristics, including gender and age.

Enhanced sleep efficiency and reduced WASO is likely to improve sleep quality by minimizing sleep fragmentation and promoting more consolidated sleep episodes. These findings suggest that RTA not only improves nocturnal physiological processes but also supports better cognitive and emotional performance during the day. The reduced REM latency under RTA is particularly noteworthy, as shortened latency to REM sleep has been associated with better mood regulation and memory consolidation. By fostering a more efficient transition into restorative sleep stages, adaptive thermal regulation may have far-reaching implications for individuals with mood disorders or cognitive impairments linked to poor sleep quality [36,37].

The efficacy of RTA in enhancing sleep architecture can be attributed to its alignment with the body’s circadian and sleep-stage-specific thermoregulatory processes. During REM sleep, the body’s thermoregulatory capacity is significantly reduced, making it more vulnerable to external temperature fluctuations [38,39]. By lowering the mattress temperature during REM sleep, RTA compensates for this vulnerability, maintaining an optimal thermal environment that supports uninterrupted REM cycles. Similarly, the return to a higher baseline temperature during non-REM sleep aligns with the body’s increased thermoregulatory activity, promoting deep sleep and recovery. The pre-wake temperature increase observed in the RTA condition likely aids in facilitating arousal and reducing sleep inertia. This mechanism mirrors natural circadian temperature rhythms, where a rise in core body temperature occurs in the early morning hours to prepare the body for wakefulness [40]. By synchronizing mattress temperature with these physiological changes, RTA enhances the natural sleep–wake cycle and promotes smoother transitions between sleep and wake states.

Despite the promising results, this study has several limitations that should be addressed in future research. First, the sample size of 25 participants, although sufficient for preliminary analysis, limits the generalizability of the findings. Larger-scale studies are needed to confirm the efficacy of RTA across diverse populations and age groups. Nevertheless, our study overcomes this limitation by employing a rigorous protocol involving polysomnographic assessments conducted on three separate occasions. This approach enhances the reliability of the findings and underscores the robustness of our methodology. Second, the short duration of this study precludes an evaluation of the long-term benefits and potential drawbacks of adaptive thermal regulation. Prolonged use of RTA and its impact on chronic sleep issues, health outcomes, and potential habituation effects remain unexplored. Another limitation is that neither skin temperature nor core body temperature was measured due to technical constraints, despite the fact that objective thermal physiology is essential for understanding the mechanisms of adaptive thermal regulation. In addition, chronotype was not formally assessed in this study, although circadian preference may influence sleep architecture. Nonetheless, the robust within-subject design of this study provides a strong foundation for understanding immediate impacts. Third, while this study employed robust PSG methods, the reliance on a single thermal mattress’s model and fixed temperature settings may not account for individual variations in thermal preferences and sensitivity. Future research should investigate customizable temperature algorithms to enhance user comfort and effectiveness. Despite this limitation, our use of AI-based, real-time adjustments represents a significant advancement over static-temperature controls. Finally, this study did not include participants with diagnosed sleep disorders, such as insomnia or obstructive sleep apnea, limiting the applicability of findings to clinical populations. Expanding the scope of RTA research to include these groups could provide valuable insights into its therapeutic potential. Nevertheless, our findings contribute valuable knowledge on the general population, serving as a critical reference for extending applications to clinical settings.

While the findings of this study highlight the benefits of adaptive thermal regulation, several avenues for future research remain. Longitudinal studies are needed to assess the sustained effects of RTA on sleep quality and its potential long-term health benefits. For instance, chronic improvements in sleep architecture could mitigate the risk of conditions such as hypertension, diabetes, and obesity, which are often associated with poor sleep quality. Additionally, exploring the application of RTA in populations with specific sleep disorders, such as insomnia, obstructive sleep apnea, or restless leg syndrome, could provide valuable insights into its therapeutic potential. Customizing thermal interventions for different demographic groups, including older adults and individuals with chronic illnesses, could further enhance the utility of RTA in promoting restorative sleep across diverse populations. Finally, advancements in wearable technology and AI-based sleep monitoring systems could enable the development of more sophisticated and user-friendly adaptive thermal regulation devices. By integrating real-time physiological data with personalized thermal adjustment algorithms, future innovations could provide tailored solutions that maximize sleep quality and overall well-being.

## 5. Conclusions

Adaptive thermal regulation represents a promising intervention for optimizing sleep quality by dynamically aligning mattress temperature with the body’s thermoregulatory needs during different sleep stages. The significant improvements in sleep architecture observed in this study underscore the potential of RTA as a non-invasive and personalized approach to enhancing restorative sleep. By addressing individual and demographic variations in thermoregulatory responses, RTA could pave the way for innovative sleep solutions that promote better health and quality of life.

## Figures and Tables

**Figure 1 healthcare-13-02521-f001:**
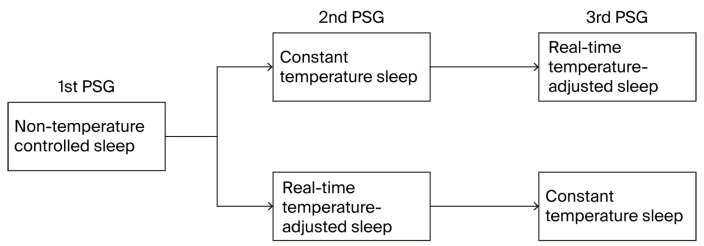
Experimental sleep conditions in a crossover design with three PSG sessions. This figure illustrates the three experimental sleep conditions in a crossover study design, where each participant underwent three separate polysomnography sessions. The conditions included non-temperature-controlled sleep (baseline without mattress intervention), constant-temperature sleep (fixed mattress temperature), and real-time temperature-adjusted sleep (dynamic adjustments based on sleep stages).

**Figure 2 healthcare-13-02521-f002:**
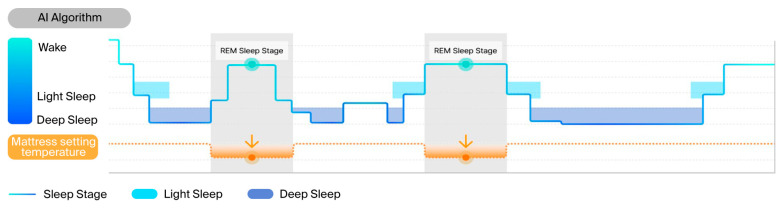
Temperature-controlled mattress protocols: baseline, CTC, and RTA conditions. This figure depicts the mattress temperature protocols used in this study. In the baseline condition, no temperature control was applied. In the CTC condition, the mattress temperature was kept constant at 33 °C. In the RTA condition, the temperature was dynamically adjusted—30 °C during REM sleep, 33 °C during non-REM sleep, and 36 °C before waking.

**Figure 3 healthcare-13-02521-f003:**
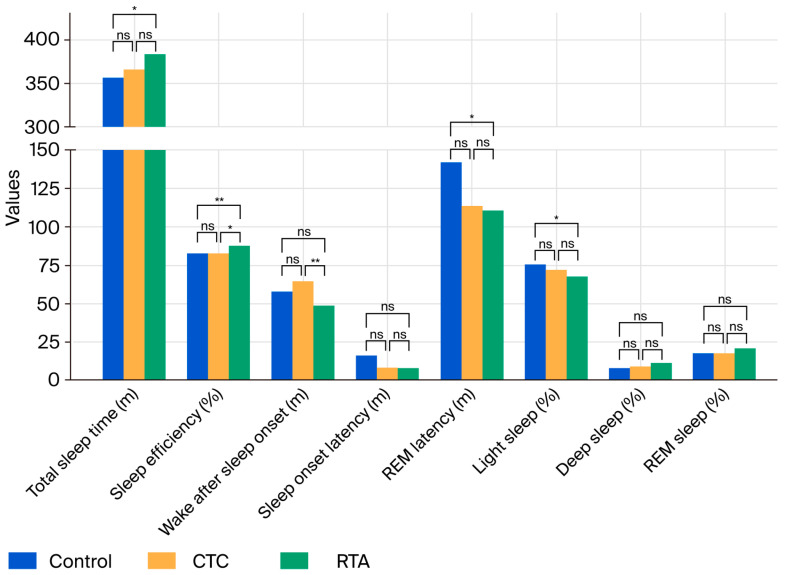
This figure displays the comparative effects of controlled temperature control (CTC) and real-time temperature adjustment (RTA) mattress settings on sleep parameters in 25 participants. The parameters include total sleep time, sleep efficiency, wake after sleep onset, sleep onset latency, REM latency, light sleep percentage, deep sleep percentage, and REM sleep percentage. Data were derived from polysomnography measurements. Bars represent mean values for each parameter under these three conditions: control, CTC, and RTA. * *p* < 0.05, ** *p* < 0.01, ns: not significant.

**Figure 4 healthcare-13-02521-f004:**
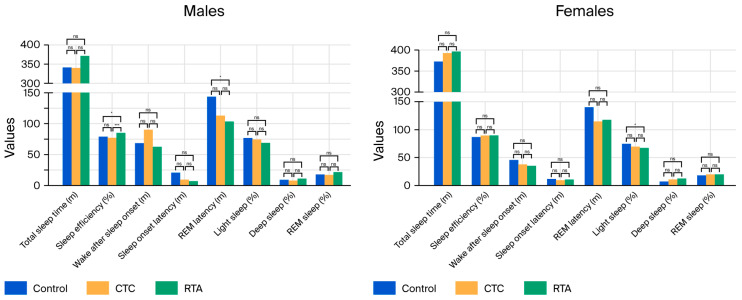
**Subgroup analysis of temperature-controlled mattress effects on sleep parameters: male vs. female results**. This figure presents the subgroup analysis of polysomnography results comparing the effects of controlled temperature control (CTC) and real-time temperature adjustment (RTA) mattress settings on sleep parameters. The left panel illustrates the results for 13 male participants, while the right panel shows the results for 12 female participants. Bars represent mean values for each condition (control, CTC, and RTA). The analysis highlights gender-specific differences in the impact of mattress temperature adjustments on sleep quality. * *p* < 0.05, *** *p* < 0.001, ns: not significant.

**Table 1 healthcare-13-02521-t001:** Effects of temperature interventions on sleep parameters.

	Repeated-Measures ANOVA Analysis				Post Hoc Analysis		
Parameters	Control	CTC	RTA	*p*-Value	Control vs. CTC	Control vs. RTA	CTC vs. RTA
Total sleep time (m)	356.2 ± 54.3	365.2 ± 67.7	383.2 ± 60.1	0.030	0.856	0.025	0.054
Sleep efficiency (%)	82.8 ± 12.3	83.1 ± 14.7	87.3 ± 13.0	0.030	0.900	0.004	0.010
Wake after sleep onset (m)	58.2 ± 41.5	64.6 ± 57.1	49.0 ± 54.9	0.067	0.326	0.237	0.041
Sleep onset latency (m)	16.3 ± 33.6	8.4 ± 7.4	8.3 ± 6.5	0.383	0.603	0.161	0.903
REM latency (m)	141.8 ± 73.7	113.5 ± 59.3	110.4 ± 60.2	0.002	0.052	0.020	0.710
Light sleep (%)	75.8 ± 8.5	72.0 ± 10.2	67.9 ± 10.1	0.002	0.341	0.011	0.264
Deep sleep (%)	8.1 ± 14.7	9.1 ± 6.2	11.4 ± 8.0	0.482	0.945	0.486	0.685
REM sleep (%)	17.7 ± 6.1	18.0 ± 5.7	20.8 ± 5.0	0.006	0.963	0.115	0.191

CTC, constant temperature control; RTA, real-time temperature adjusted; REM, rapid eye movement. Values in the post hoc analysis indicate *p*-values.

**Table 2 healthcare-13-02521-t002:** Subgroup analysis for the effects of temperature interventions.

	Repeated-Measures ANOVA Analysis				Post Hoc Analysis		
Parameters	Control	CTC	RTA	*p*-Value	Control vs. CTC	Control vs. RTA	CTC vs. RTA
Males (*n* = 13)							
Total sleep time (m)	341.1 ± 54.7	339.3 ± 79.4	370.9 ± 76.0	0.089	0.998	0.564	0.526
Sleep efficiency (%)	79.3 ± 13.1	77.2 ± 17.1	85.1 ± 16.7	0.001	0.497	0.017	0.001
Wake after sleep onset (m)	68.5 ± 42.5	90.2 ± 63.6	62.1 ± 69.6	0.062	0.650	0.963	0.489
Sleep onset latency (m)	21.0 ± 44.2	8.1 ± 8.9	6.8 ± 6.4	0.066	0.799	0.068	0.273
REM latency (m)	143.3 ± 80.8	112.3 ± 71.9	103.6 ± 76.4	0.001	0.066	0.018	0.297
Light sleep (%)	76.5 ± 8.6	74.1 ± 12.3	68.9 ± 11.9	0.170	0.861	0.226	0.485
Deep sleep (%)	9.1 ± 18.9	7.6 ± 6.6	10.7 ± 8.1	0.859	0.954	0.950	0.823
REM sleep (%)	17.5 ± 7.5	16.7 ± 6.7	21.3 ± 5.2	0.010	0.945	0.348	0.209
Females (*n* = 12)							
Total sleep time (m)	372.6 ± 48.8	393.3 ± 34.7	396.6 ± 30.4	0.208	0.432	0.326	0.978
Sleep efficiency (%)	86.6 ± 10.1	89.4 ± 7.7	89.7 ± 6.2	0.490	0.701	0.653	0.997
Wake after sleep onset (m)	45.3 ± 36.8	37.6 ± 31.3	34.8 ± 25.8	0.627	0.838	0.722	0.977
Sleep onset latency (m)	11.8 ± 13.9	9.2 ± 5.4	10.3 ± 6.2	0.646	0.795	0.920	0.964
REM latency (m)	140.2 ± 65.1	114.7 ± 41.6	117.7 ± 33.7	0.297	0.380	0.339	0.505
Light sleep (%)	75.1 ± 8.2	69.8 ± 6.7	66.7 ± 7.5	0.001	0.238	0.035	0.065
Deep sleep (%)	7.1 ± 7.9	10.6 ± 5.4	12.3 ± 7.8	0.011	0.489	0.220	0.848
REM sleep (%)	17.8 ± 4.1	19.6 ± 3.9	20.3 ± 4.7	0.249	0.598	0.364	0.915

CTC, constant temperature control; RTA, real-time temperature adjusted; REM, rapid eye movement. Values in the post hoc analysis indicate *p*-values.

## Data Availability

Data are contained within the article.

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
