# Peer review of "Polysomnographic Evidence of Enhanced Sleep Quality with Adaptive Thermal Regulation"

_healthcare, 2025, doi:10.3390/healthcare13192521_

Round 1
Reviewer 1 Report
Comments and Suggestions for Authors
x
This is an interesting paper studying sleep changes under different temperature conditions with a heated mattress according to REM or NREM sleep. There are only few studies that have combined temperature changes and sleep sleep parameters without measuring surface or core body temperature.
The three study conditions were well chosen.
- How exactly was the AI regulation of the thermal mattress? Has this method been validated? Were the temperature changes performed according to the Ai algorithm of breathing sounds or the real time PSG changes? Was PSG performed in each of the conditions?
- Were probands instructed not to perform sports or take hot meals some hours before they were put to sleep? These activities could raise core body temperature.
- How long did probands adapt to the ambient temperature prior to sleep.
- Did authors assess for late and early type sleepers to rule out circadian effects of sleep stage changes.
- Did participants know the different study conditions to which they were randomly assigned? (ie. that temperature would be changed etc.)
- Were probands asked to keep their hands under the blanket to allow the loss of temperature according to the mattress temperature?
- What was the mattress temperature under natural sleep conditions?
- Were there any measures of skin temperature to make sure that the peripheral vessels dilated? Was core temperature affected by the different conditions during real-time temperature adjusted to sleep to make sure that changes are related to objective parameters of temperature?
- Were any differences in sleep stage transition observed depending on the body positions throughout the night.
- Considering the missing change of sleep latency among the conditions: How long before starting recording were patients laid on the mattress?
- In the graphs significant changes should be marked by asterixes.
- Discussion: the marked changes in REM latency and REM% should be differentiated for m and f.
- The same goes fort he increase of deep sleep. To interpret the tables it would be good to show standard deviations
Over all this is a very interesting study which could after some clarifications stimulate new studies in patients with sleep disorders like insomnia or with misalignment.
Author Response
Q1:How exactly was the AI regulation of the thermal mattress? Has this method been validated? Were the temperature changes performed according to the Ai algorithm of breathing sounds or the real time PSG changes? Was PSG performed in each of the conditions?
A1:
We appreciate the reviewer’s insightful questions. The thermal mattress was regulated using an AI-based application that continuously recorded participants’ nocturnal breathing sounds and predicted sleep stages in real time. Mattress temperature was dynamically adjusted according to these AI-derived sleep stage transitions, not by simultaneous PSG scoring.
Regarding validation, the sound-based AI model has been rigorously evaluated in a recent clinical study with synchronized PSG recordings as the gold standard (“Evaluation of sound-based sleep stage prediction in shared sleeping settings,” Sleep Medicine, 2025). This study demonstrated that the algorithm achieved macro F1 score of 0.77, indicating substantial agreement with PSG.
In our study, full-night level 1 PSG was indeed performed in each condition to ensure standardized evaluation and to provide independent verification of sleep architecture. However, the adaptive thermal regulation itself was driven solely by the AI algorithm’s real-time sleep stage prediction from breathing sounds.
Q2: Were probands instructed not to perform sports or take hot meals some hours before they were put to sleep? These activities could raise core body temperature.
A2:
We appreciate the reviewer’s valuable suggestion. To minimize confounding factors that could affect core body temperature, participants were instructed to refrain from caffeine intake (coffee, tea, and other caffeinated beverages) on the day of each PSG session. In addition, they were asked to avoid meals, alcohol, and exercise for at least 6 hours prior to the scheduled recording. We have now explicitly stated these instructions in the revised Methods section.
Q3: How long did probands adapt to the ambient temperature prior to sleep.
A3: We appreciate the reviewer’s attention to this important detail. Participants were given an adaptation period of approximately 1 hour in the controlled sleep environment (18–20 °C, 50–55% humidity) before lights off. This information has now been explicitly added to the Methods section.
Q4: Did authors assess for late and early type sleepers to rule out circadian effects of sleep stage changes.
A4: We sincerely thank the reviewer for this thoughtful suggestion. Chronotype was not formally assessed in this pilot study. We fully agree that circadian preference could influence sleep stage dynamics, and we consider this an excellent idea for future research. We have now acknowledged this as a limitation in the Discussion section and suggested that subsequent studies stratify participants according to chronotype to better control for inter-individual variability.
Q5: Did participants know the different study conditions to which they were randomly assigned? (ie. that temperature would be changed etc.)
A5: We thank the reviewer for highlighting this important point. Participants were informed that mattress temperature might vary across nights; however, they were blinded to the specific protocols and the timing of temperature changes. This ensured that participants remained unaware of the study condition to which they were assigned on a given night. We have now clarified this blinding procedure in the Methods section.
Q6: Were probands asked to keep their hands under the blanket to allow the loss of temperature according to the mattress temperature?
A6: We are grateful for this thoughtful suggestion. Participants were instructed to maintain their usual sleeping posture, and no specific instructions were given regarding hand placement. We agree that hand positioning could influence heat exchange with the mattress, and this is indeed a valuable idea to consider for future studies.
Q7: What was the mattress temperature under natural sleep conditions?
A7: We thank the reviewer for pointing out this important detail. In the Control (natural sleep) condition, the mattress heating function was not activated, and no temperature modulation was applied. Temperature regulation of the mattress was conducted only under the CTC and RTA conditions. We have clarified this in the revised Methods section.
Q8: Were there any measures of skin temperature to make sure that the peripheral vessels dilated? Was core temperature affected by the different conditions during real-time temperature adjusted to sleep to make sure that changes are related to objective parameters of temperature?
A8: We appreciate this important comment. Due to technical limitations, neither skin temperature nor core body temperature was measured in this study. We fully agree that objective physiological markers of thermoregulation are crucial to better understand the mechanisms underlying adaptive thermal regulation. We have therefore acknowledged this as a limitation and suggested that future studies incorporate both skin and core body temperature measurements to provide more comprehensive insights.
Q9: Were any differences in sleep stage transition observed depending on the body positions throughout the night.
A9: We sincerely thank the reviewer for this valuable idea. Body position was not separately recorded or analyzed in the present study. We agree that posture during sleep may influence sleep stage transitions, and we consider this an important factor to include in future work.
Q10: Considering the missing change of sleep latency among the conditions: How long before starting recording were patients laid on the mattress?
A10: Participants were asked to lie on the mattress approximately 1 hour before lights off to ensure sufficient adaptation. This information has been added to the revised Methods section.
Q11: In the graphs significant changes should be marked by asterixes.
A11: We thank the reviewer for this helpful suggestion. All figures have been revised, and statistically significant changes are now indicated with asterisks.
Q12: Discussion: the marked changes in REM latency and REM% should be differentiated for m and f.
A12: We sincerely thank the reviewer for this valuable suggestion. In our analyses, both male and female participants showed similar trends in REM latency, REM%, and deep sleep percentage. Therefore, we did not provide separate sex-specific discussions in the revised manuscript. We agree, however, that stratified analyses may provide further insights in larger studies with greater statistical power
Q13: The same goes fort he increase of deep sleep. To interpret the tables it would be good to show standard deviations
A13: We thank the reviewer for this constructive suggestion. All tables have been revised to include standard deviations alongside mean values.
Reviewer 2 Report
Comments and Suggestions for Authors
This study examined the effect of bed temperature on sleep quality by comparing three conditions: natural sleep, a constant temperature of 33 °C, and dynamic temperature adjustment according to sleep stages. The results showed that dynamic temperature control increased total sleep time and efficiency while shortening REM latency.
It is recommended that the following points be revised:
-
Although the manuscript reports different effects between male and female participants, it does not indicate whether the sample sizes for these subgroups were statistically sufficient. A power analysis should be provided.
-
The study relies solely on short-term PSG sessions. The long-term effects of dynamic temperature control on sleep architecture and health outcomes have not been evaluated. A discussion on this should be included.
-
Inclusion and exclusion criteria should be described in detail, and information about participants’ history of sleep disorders should be provided.
-
The rationale for decreasing the temperature by exactly 3 °C during REM sleep should be explained and supported with references.
-
In some parameters, repeated-measures ANOVA indicated significant differences, whereas post-hoc analyses did not reveal significant pairwise differences. This discrepancy should be discussed.
This study examined the effect of bed temperature on sleep quality by comparing three conditions: natural sleep, a constant temperature of 33 °C, and dynamic temperature adjustment according to sleep stages. The results showed that dynamic temperature control increased total sleep time and efficiency while shortening REM latency.
It is recommended that the following points be revised:
-
Although the manuscript reports different effects between male and female participants, it does not indicate whether the sample sizes for these subgroups were statistically sufficient. A power analysis should be provided.
-
The study relies solely on short-term PSG sessions. The long-term effects of dynamic temperature control on sleep architecture and health outcomes have not been evaluated. A discussion on this should be included.
-
Inclusion and exclusion criteria should be described in detail, and information about participants’ history of sleep disorders should be provided.
-
The rationale for decreasing the temperature by exactly 3 °C during REM sleep should be explained and supported with references.
-
In some parameters, repeated-measures ANOVA indicated significant differences, whereas post-hoc analyses did not reveal significant pairwise differences. This discrepancy should be discussed.
Author Response
Q1: Although the manuscript reports different effects between male and female participants, it does not indicate whether the sample sizes for these subgroups were statistically sufficient. A power analysis should be provided.
A1: We thank the reviewer for this important comment. A power analysis indicated that the overall sample size of 25 participants was sufficient to detect within-subject effects using repeated-measures ANOVA. However, when divided into male and female subgroups, the statistical power was inevitably limited due to the smaller numbers in each group. We have now acknowledged this limitation in the Discussion, noting that sex-specific findings should be interpreted with caution and that larger-scale studies are required to validate these subgroup effects.
We appreciate the reviewer’s request for a power analysis. For our primary within-subject comparisons, we conducted an a priori/achieved power check using G*Power with the following settings for a repeated-measures ANOVA (within factors):
- Test family: F tests → Statistical test: ANOVA: Repeated measures, within factors
- Number of measurements (levels): 3 (Control, CTC, RTA)
- Correlation among repeated measures (ρ): 0.3–0.5 (assumed)
- Nonsphericity correction (ε): 0.75–1.00 (assumed)
- α = 0.05, total sample size n = 25
Under conventional effect sizes, the achieved power is adequate for the overall within-subject design: for a medium effect (Cohen’s f = 0.25), the estimated power is approximately 0.80–0.88 across the above assumptions; for a large effect (f = 0.40), power exceeds 0.95. Conversely, for a small effect (f = 0.10–0.15), power is limited (~0.20–0.55). These ranges reflect typical values when ρ and ε vary within reasonable bounds.
However, sex-stratified subgroup analyses are under-powered given the smaller per-group sample sizes. With n per subgroup reduced (e.g., ~12–13), the estimated power for f = 0.25 drops to roughly ~0.45–0.65, depending on ρ and ε. We have therefore added a statement to the Discussion noting that sex-specific findings should be interpreted with caution and confirmed in larger samples.
(Optional concise line you can paste as-is)
“In G*Power (RM ANOVA, within factors; 3 levels; α = 0.05; ρ = 0.3–0.5; ε = 0.75–1.00), the overall sample (n = 25) affords ~0.80–0.88 power for a medium effect (f = 0.25) and >0.95 for a large effect (f = 0.40); sex-specific subgroups are under-powered.”
Q2: The study relies solely on short-term PSG sessions. The long-term effects of dynamic temperature control on sleep architecture and health outcomes have not been evaluated. A discussion on this should be included.
A2: We sincerely thank the reviewer for this important comment. Conducting three full-night level 1 PSG sessions per participant was already highly demanding in terms of both participant burden and study logistics. As such, this study was limited to short-term comparisons. We have acknowledged this limitation in the Discussion. At the same time, we fully agree that the long-term effects of adaptive thermal regulation remain to be elucidated, and we are planning follow-up studies, including prospective trials with longer-term monitoring such as one-month follow-up designs, to further address this important question.
Q3: Inclusion and exclusion criteria should be described in detail, and information about participants’ history of sleep disorders should be provided.
A3: We thank the reviewer for this valuable comment. We have expanded the Methods section to provide more detailed descriptions of the inclusion and exclusion criteria. Participants with a history of clinically diagnosed sleep disorders were classified as exclusion cases and were not included in this study.
Q4: The rationale for decreasing the temperature by exactly 3 °C during REM sleep should be explained and supported with references.
A4: We thank the reviewer for this insightful comment. There is no direct physiological evidence supporting the specific value of a 3 °C decrease during REM sleep. However, our rationale for this setting was as follows: the initial mattress temperature of 33 °C was selected because it represents a commonly perceived “warm and comfortable” setting for winter conditions in Korea. Since REM sleep is characterized by impaired thermoregulation, maintaining this warmer setting was considered potentially disruptive, as it may increase thermal discomfort and interfere with REM expression. Therefore, the mattress temperature was lowered by 3 °C specifically during REM stages, while it reverted to the baseline 33 °C during non-REM stages. The rationale for lowering temperature in REM sleep has been supported by previous studies on altered thermoregulatory control during REM sleep, which are now cited in the revised manuscript.
Q5: In some parameters, repeated-measures ANOVA indicated significant differences, whereas post-hoc analyses did not reveal significant pairwise differences. This discrepancy should be discussed.
A5: We thank the reviewer for this important observation. The repeated-measures ANOVA allowed us to detect overall effects of the RTA intervention across conditions. Post-hoc analyses, on the other hand, were conducted to explore pairwise differences among Control, CTC, and RTA. It is possible for the omnibus ANOVA to show significance, while the post-hoc comparisons fail to reach significance after adjustment for multiple testing. This reflects the greater stringency of pairwise comparisons and the reduced power following correction. We have now clarified this point in the Discussion to explain why overall effects were observed in the ANOVA but not consistently confirmed in post-hoc tests.
Reviewer 3 Report
Comments and Suggestions for Authors
This manuscript titled “Polysomnographic Evidence of Enhanced Sleep Quality with Adaptive Thermal Regulation” evaluates a temperature-controlled mattress that uses an AI-trained model to dynamically adjust surface temperature based on sleep stages. Unlike previous devices that only adjust ambient temperature, this system applies a real-time adjustable temperature protocol (RTA). The authors report that RTA increases total sleep time and sleep efficiency, though not REM sleep. Stratified analysis revealed sex-specific effects: males showed increased sleep efficiency, while females exhibited a higher percentage of deep sleep.
Overall, this study addresses an interesting question and the technology has potential implications for improving sleep quality. However, the manuscript lacks sufficient rigor and supporting evidence for its conclusions.
Major Concerns:
-
The authors lowered mattress temperature by 3°C during REM sleep compared to non-REM stages. However, it is well established that body temperature is tightly regulated (and lower) during non-REM sleep but less so during REM. The rationale for applying lower temperature specifically during REM—and for raising temperature before awakening—requires further justification, supported by relevant physiological evidence.
-
The study assumes that mattress surface temperature is the dominant factor influencing skin and core body temperature. Please provide justification and, if available, supporting evidence that mattress-controlled skin temperature outweighs the effects of overall ambient room temperature.
-
The sample size (<30 participants) is underpowered, which may explain why many comparisons are statistically insignificant. More critically, the crossover design (participants sequentially undergoing CTC and RTA) raises concerns about carryover or long-term adaptation effects. Did the authors test whether treatment order influenced outcomes? Have comparisons of the same condition across both groups been performed to validate reproducibility?
-
The AI-driven mattress determines sleep stage based on breathing signals. However, the accuracy of this classification is not validated. The authors should directly compare mattress-derived staging with PSG (gold standard) data to establish reliability.
-
Minor Concerns: figures (3-4) need error bars and figure 5 is missing.
Author Response
Q1: The authors lowered mattress temperature by 3°C during REM sleep compared to non-REM stages. However, it is well established that body temperature is tightly regulated (and lower) during non-REM sleep but less so during REM. The rationale for applying lower temperature specifically during REM—and for raising temperature before awakening—requires further justification, supported by relevant physiological evidence.
A1: We thank the reviewer for this insightful comment. There is no direct physiological evidence supporting the specific value of a 3 °C decrease during REM sleep. However, our rationale for this setting was as follows: the initial mattress temperature of 33 °C was selected because it represents a commonly perceived “warm and comfortable” setting for winter conditions in Korea. Since REM sleep is characterized by impaired thermoregulation, maintaining this warmer setting was considered potentially disruptive, as it may increase thermal discomfort and interfere with REM expression. Therefore, the mattress temperature was lowered by 3 °C specifically during REM stages, while it reverted to the baseline 33 °C during non-REM stages. The rationale for lowering temperature in REM sleep has been supported by previous studies on altered thermoregulatory control during REM sleep, which are now cited in the revised manuscript.
In addition, the rationale for raising mattress temperature before awakening was based on circadian physiology. It is well established that core body temperature naturally rises toward the end of the sleep period, serving as a physiological signal for awakening. By slightly increasing mattress temperature prior to scheduled wake time, our protocol aimed to mimic this endogenous circadian rise in body temperature and thereby support a more natural transition from sleep to wakefulness【ref: Okamoto et al., 2008, 2012; 】.
Q2: The study assumes that mattress surface temperature is the dominant factor influencing skin and core body temperature. Please provide justification and, if available, supporting evidence that mattress-controlled skin temperature outweighs the effects of overall ambient room temperature.
A2: We thank the reviewer for this important point. Our study protocol assumed that mattress surface temperature is a dominant factor in thermoregulation during sleep, given that it is in direct contact with a large portion of the body surface. Prior research has demonstrated that proximal skin temperature influenced by bedding or mattress conditions has a stronger effect on sleep initiation and maintenance than changes in ambient room temperature alone (e.g., Kräuchi & Wirz-Justice, 2001; Okamoto-Mizuno & Mizuno, 2012). Thus, we considered mattress-controlled skin temperature to be the primary determinant in our thermal intervention.
We acknowledge, however, that we did not measure skin or core body temperatures in the present study. This is now noted as a limitation in the Discussion, and we have emphasized the importance of including objective thermal physiology measurements in future work to validate this assumption.
Q3: The sample size (<30 participants) is underpowered, which may explain why many comparisons are statistically insignificant. More critically, the crossover design (participants sequentially undergoing CTC and RTA) raises concerns about carryover or long-term adaptation effects. Did the authors test whether treatment order influenced outcomes? Have comparisons of the same condition across both groups been performed to validate reproducibility?
A3: We sincerely thank the reviewer for this important comment. Regarding sample size, a priori/achieved power analysis (RM ANOVA, within-subject, 3 conditions; α = 0.05; n = 25) indicated that our overall sample was adequate to detect medium effect sizes (power ≈ 0.80–0.88 for f = 0.25) and sufficient for larger effects (power > 0.95 for f = 0.40). However, we acknowledge that subgroup analyses, such as sex-specific comparisons, were underpowered and should be interpreted with caution. We have noted this limitation in the Discussion and emphasized the need for larger-scale studies to validate these findings.
With respect to the crossover design, all participants first underwent the Control condition, and the two intervention conditions (CTC and RTA) were then administered in a randomized and blinded order. In addition, a 5-day washout interval (“window visit”) was implemented between conditions to minimize potential carryover or long-term adaptation effects. We have clarified these design features in the revised Methods section.
Nonetheless, we fully agree with the reviewer that future studies with larger sample sizes and independent replication are warranted to further examine reproducibility and to address possible long-term adaptation effects.
Q4: The AI-driven mattress determines sleep stage based on breathing signals. However, the accuracy of this classification is not validated. The authors should directly compare mattress-derived staging with PSG (gold standard) data to establish reliability.
A4: We thank the reviewer for this valuable comment. The AI algorithm used in our study was not newly validated within this dataset; rather, it is based on a previously developed and independently validated model. In that prior study, the algorithm was directly compared with simultaneous PSG recordings and demonstrated a macro F1 score of approximately 0.76, indicating substantial agreement with PSG-based staging(inside ref 39,40). We have now cited this validation work in the revised manuscript to clarify that the classification method has been previously established as reliable. In our study, mattress regulation was based on these AI-derived stage predictions, while PSG served as the gold standard outcome measure for analysis.
Q5: Minor Concerns: figures (3-4) need error bars and figure 5 is missing.
A5: We thank the reviewer for this helpful suggestion. Figures 3 and 4 have been revised to include error bars. Regarding Figure 5, the content has been integrated into Figure 4 in the revised manuscript to improve clarity and avoid redundancy.
Round 2
Reviewer 1 Report
Comments and Suggestions for Authors
All the reviewers comments were sufficiently answered. Authors did a great job
Author Response
Thank you very much for your thoughtful review and positive feedback.
We truly appreciate your careful reading and constructive comments.
In particular, the points you raised regarding areas that could be further explored will be carefully considered in our future studies.
We will make sure to design follow-up research that addresses these limitations and expands on the current findings.
Reviewer 3 Report
Comments and Suggestions for Authors
The most of my questions are addressed by the authors, and I have no further questions for its publication.
Author Response

(The authors gave the same response as above.)
